# Antimicrobial Peptide LCN2 Inhibited Uropathogenic *Escherichia coli* Infection in Bladder Cells in a High-Glucose Environment through JAK/STAT Signaling Pathway

**DOI:** 10.3390/ijms232415763

**Published:** 2022-12-12

**Authors:** Pei-Chi Chen, Chen-Hsun Ho, Chia-Kwung Fan, Shih-Ping Liu, Po-Ching Cheng

**Affiliations:** 1Division of Endocrinology and Metabolism, Shin Kong Wu Ho-Su Memorial Hospital, Taipei 11101, Taiwan; 2School of Medicine, College of Medicine, Fu Jen Catholic University, New Taipei City 24205, Taiwan; 3Division of Urology, Department of Surgery, Shin Kong Wu Ho-Su Memorial Hospital, Taipei 11101, Taiwan; 4Department of Molecular Parasitology and Tropical Diseases, School of Medicine, College of Medicine, Taipei Medical University, Taipei 11031, Taiwan; 5Center for International Tropical Medicine, College of Medicine, Taipei Medical University, Taipei 11031, Taiwan; 6Department of Urology, National Taiwan University Hospital and College of Medicine, Taipei 10002, Taiwan

**Keywords:** bladder cells, antimicrobial peptide, LCN2, JAK/STAT, uropathogenic *Escherichia coli*

## Abstract

JAK/STAT plays a key role in regulating uropathogenic *Escherichia coli* (UPEC) infection in urothelial cells, probably via antimicrobial peptide (AMP) production, in diabetic patients with urinary tract infections. Whether multiple pathways regulate AMPs, especially lipid-carrying protein-2 (LCN2), to achieve a vital effect is unknown. We investigated the effects of an LCN2 pretreatment on the regulation of the JAK/STAT pathway in a high-glucose environment using a bladder cell model with GFP-UPEC and phycoerythrin-labeled TLR-4, STAT1, and STAT3. Pretreatment with 5 or 25 μg/mL LCN2 for 24 h dose-dependently suppressed UPEC infections in bladder cells. TLR-4, STAT1, and STAT3 expression were dose-dependently downregulated after LCN2 pretreatment. The LCN2-mediated alleviation of UPEC infection in a high-glucose environment downregulated TLR-4 and the JAK/STAT transduction pathway and decreased the UPEC-induced secretion of exogenous inflammatory interleukin (IL)-6 and IL-8. Our study provides evidence that LCN2 can alleviate UPEC infection in bladder epithelial cells by decreasing JAK/STAT pathway activation in a high-glucose environment. LCN2 dose-dependently inhibits UPEC infection via TLR-4 expression and JAK/STAT pathway modulation. These findings may provide a rationale for targeting LCN2/TLR-4/JAK/STAT regulation in bacterial cystitis treatment. Further studies should explore specific mechanisms by which the LCN2, TLR-4, and JAK/STAT pathways participate in UPEC-induced inflammation to facilitate the development of effective therapies for cystitis.

## 1. Introduction

Diabetes increases the risk of infection in patients and is associated with many complications, the most common being bacterial infection of the urinary tract. The incidence of urinary tract infection (UTI) in diabetes mellitus (DM) patients is 10 times higher, and the risk of a UTI causing sepsis and mortality is 4 to 5 times greater than in nondiabetic patients [1]. In addition, long-term hyperglycemia in patients with DM can weaken the immune system [2,3]. The most common pathogenic bacterium for UTIs in diabetic patients is *Escherichia coli*, followed by *Klebsiella pneumoniae*, *Enterococci*, and other bacteria [4,5,6,7]. Women belong to the high-risk group for UTIs. Studies have shown that up to one third of women may experience UTIs during their lifetime [3,8,9,10]. Previous studies have found that the adhesion of *E. coli* cilia to the urinary epithelium is higher in DM patients and positively correlates with the level of glycosylated hemoglobin [11]. The glycoprotein on the cilia of *E. coli* also binds more tightly to the glycoprotein receptor of the urinary tract epithelial cells of patients with DM, thus increasing the chance of infection [1]. Studies have found that type 2 diabetic mice with hyperglycemia have significantly increased UTI sensitivity and decreased uropathogenic *Escherichia coli* (UPEC) clearance, which cause more serious infections [12]. Therefore, it is essential to understand the mechanism by which the urinary tract epithelial tissue in the diabetic state is invaded by pathogens.

Antimicrobial peptides (AMPs) are an important part of the urothelial innate immune system and serve as natural antibiotics to prevent pathogens from invading the urinary tract. AMPs can be developed as new UTI therapeutic drugs [13]. AMPs are mainly produced by intercalated cells (ICs) in the kidney and bladder to help the antibacterial immune defense of urothelial tissues [13]. Mice with reduced ICs have a significant increase in urine and bladder bacteria upon infection with UPEC [14]. Current research shows that an induction of AMP production may activate insulin and insulin-mediated phosphatidylinositol-3-kinase (PI3K)/protein kinase B (AKT) transduction pathways and consequently enable urinary tract epithelial cells to avoid UPEC invasion and infection [15]. The concentrations of AMPs such as Rnase4 and lipid-carrying protein-2 (LCN2) peptides in the urine of young people suffering from T2DM show significant negative correlations with blood sugar control [16]. Murtha et al. further proved that insulin can regulate AMPs expressed by ICs through the PI3K/AKT pathway and reduce the concentration of Rnase4 and Lcn2 in the urine to decelerate the clearance of UPEC; this phenomenon may make insulin receptor (IR) knockout mice more likely to experience UTI [17]. UPEC itself can also inhibit PI3K/AKT activity and the downstream AMP production, as the activation of the PI3K/AKT transduction pathway also affects cell survival, growth, proliferation, migration, and other physiological functions [18]. Accumulating evidence suggests that LCN2 is involved in insulin resistance and glucose homeostasis. LCN2 regulates insulin sensitivity and glucose metabolism by inhibiting 5′-AMP-activated protein kinase (AMPK) activity and regulating FoxO1 and its downstream genes phosphoenolpyruvate carboxykinase (PEPCK)/glucose-6-phosphatase (G6P), which modulate hepatic gluconeogenesis [19]. However, the immunomechanism of cellular regulation is not mediated via a single pathway. We recently demonstrated that insulin downregulates the infection of UPEC in bladder cells in a high-glucose environment via the Janus kinase (JAK)/signal transducer and activator of transcription (STAT) signaling pathway [20]. Whether JAK/STAT signaling transduction pathways are involved in the regulation of AMPs, especially LCN2 production, to achieve a vital effect in UTI defense was our focus.

JAK aggregates and binds to the cognate ligand regions of various receptors in the cell. Activated JAK phosphorylates STAT, which forms a dimer and enters the nucleus to regulate gene transcription. Depending on the cytokines or growth factors, diverse and highly specific JAK and STAT proteins are expressed to regulate and maintain basic biological processes, including apoptosis, proliferation, immunity, and inflammation [21]. The interleukin 6 (IL6)/JAK2/STAT3 signal transduction in the skeletal muscle of T2D patients plays a pathogenic role in the process of insulin resistance. The increase in phosphorylated JAK2 and STAT3 levels is accompanied by glucose tolerance loss or T2D [22,23]. STAT3 also induces the production of Toll-like receptor (TLR)-4, resulting in the generation of proinflammatory cytokines and insulin resistance in myoblasts [23]. It is worth noting that TLR-4 greatly increases inflammation in T2D patients. Hyperglycemia and free fatty acids jointly promote oxidation and the TLR-4–mediated development of insulin resistance and endothelial dysfunction [24]. In addition, the expressions of MyD88 and nuclear factor kappa B (NF-κB), the downstream factors of TLR-4, increase in cells exposed to high glucose levels, which may lead to the secretion of the inflammatory factor IL-1β [25]. Previous studies in our laboratory also confirmed that an increase in the sugar concentration enhances UPEC-induced bladder cell inflammation. These responses were found to regulate the increase in UPEC infection of bladder cells by upregulating the mechanism of TLR-4 participation in the coordinated expansion of the media and JAK/STAT1 message transmission pathways [26]. We also showed that testosterone can effectively suppress the activation of the JAK/STAT1 signaling pathway, thereby suppressing UPEC’s invasion of prostate cells and inducing an inflammatory response [27]. In the latest study, we found that using another antimicrobial peptide, RNase 7, can effectively inhibit the UPEC-caused infection and inflammation in bladder cells, which is also regulated by the JAK/STAT signaling pathway [28]. We hypothesize that it may also be related to the regulation of LCN2 production in diabetic patients during UTI. The present study provides the possible therapeutic strategy of increasing endogenous AMP production by regulating the JAK/STAT pathway, which can be applied to prevent diabetic UTI or other DM-related infections.

## 2. Results

### 2.1. LCN2 Suppresses UPEC Infections in Bladder Cells in a High-Glucose Environment

We used a colony assay to detect the antibacterial effects of LCN2 on UPEC in bladder cells in a high-glucose environment. As shown in Figure 1, treating SV-HUC-1 cells with 15 mM glucose enhanced UPEC infection more than the general infections (*p* < 0.05). Although, a 1 μg/mL LCN2 pretreatment showed no beneficial effects, pretreating with 5 or 25 μg/mL LCN2 for 24 h significantly and dose-dependently suppressed UPEC infections in SV-HUC-1 cells compared to a 15 mM glucose treatment (*p* < 0.05) (Figure 1A,B). Therefore, we demonstrate that LCN2 can dose-dependently suppress the ability of UPEC infections in bladder cells in a high-glucose environment.

### 2.2. LCN2 Suppresses UPEC Infections and Downregulates the Expression of STAT1/3 and TLR-4 in SV-HUC-1 Cells

In order to clarify the mechanism by which LCN2 suppresses UPEC infections in bladder cells in a high-glucose environment, SV-HUC-1 cells pretreated with different concentrations of LCN2 were infected with GFP-UPEC, and the expression of specific cell markers, including STAT1, STAT3, and TLR-4, were assessed using fluorescence microscopy. Pretreatment with 5 or 25 μg/mL LCN2 significantly suppressed UPEC infection relative to the 15 mM glucose treatment (*p* < 0.001) (Figure 2A,B and Figure 3A,B). LCN2 pretreatment also downregulated STAT1 and TLR-4 expression in bladder cells in a dose-dependent manner (Figure 2A,B and Figure 4A,B) (*p* < 0.001 and *p* < 0.01, respectively, compared to the 15 mM glucose treatment). A STAT3 expression analysis showed similar results, although pretreatment with LCN2 only decreased the expression of STAT3 in the bladder cells at the 25 μg/mL concentration (*p* < 0.001 compared to the 15 mM glucose treatment) (Figure 3A,B). Notably, the expressions of STAT1, STAT3, and TLR-4 in bladder cells were even lower in the 25 μg/mL LCN2-treated group than in the UPEC-infected group (*p* < 0.05) (Figure 2, Figure 3 and Figure 4). Our results show that the AMP LCN2 not only suppresses UPEC infections but also downregulates the expression of STAT1, STAT3, and TLR-4 in bladder cells.

### 2.3. LCN2 Reduces the UPEC-Induced Exogenous Inflammatory Cytokine Secretion

In order to understand the effects of LCN2 on the UPEC-induced inflammatory responses in bladder cell lines, we investigated whether LCN2 reduces the exogenous secretion of UPEC-induced inflammatory cytokines in SV-HUC-1 cells. The CBA analysis of supernatants from cell cultures pretreated with different doses of LCN2 showed inhibitory effects on UPEC-infection-stimulated inflammation (Figure 5). After LCN2 pretreatment for 24 h, the UPEC-infection-induced secretion of both extracellular IL-8 and IL-6 significantly were reduced in all LCN2-treated groups (*p* < 0.0001 compared to the 15 mM glucose treatment). However, there was no significant difference due to the low value of extracellular IL-1β expression. In addition, there was no difference in the IL-10 secretion level between the LCN2-treated groups at all doses, the positive control group, and the 15 mM glucose-treated group.

### 2.4. LCN2-Mediated Suppression in UPEC Infection and Inflammatory Responses in Bladder Cells Is Related to Downregulation of the JAK/STAT Pathway and TLR-4 Expression

In order to verify whether LCN2 suppressed UPEC infection in bladder cells via TLR-4 activation and the JAK/STAT pathway in a high-glucose environment, the expressions of related proteins were evaluated. Figure 6 reveals the expressions of all proteins in UPEC-infected cells after pretreatment with different doses of LCN2, such as TLR-4, IL-6, and IFN-γ as well as JAK1/2, STAT1/3, and SOCS3 of the JAK/STAT signaling pathway. Our results show that the expressions of TLR-4, IL-6, IFN-γ, JAK1/2, STAT1/3, and phosphorylated STAT1/3 were induced after UPEC infection and were all significantly enhanced following the 15 mM glucose treatment (*p* < 0.05 compared to control). In addition, the expression of a STAT inhibitor, SOCS3, was slightly induced by the 25 μg/mL LCN2 pretreatment in UPEC-infected cells and showed a reverse trend compared to the regulation of the JAK-STAT signaling pathway. TLR-4 showed similar results in the 5 and 25 μg/mL LCN2 pretreatment groups (*p* < 0.01 compared to the 15 mM glucose-treated group), while the cytokines IL-6 and IFN-γ only significantly differed in the 25 μg/mL LCN2-pretreated group (*p* < 0.05 compared to the 15 mM glucose-treated group). Taken together, the expression levels were consistent for TLR-4, JAK/STAT-related transduction factors, and inflammation-related IL-6 and IFN-γ cytokines. Here, we demonstrated that LCN2 is important in the downregulation of inflammatory responses in UPEC-infected bladder cells through suppressing the JAK/STAT signaling pathway, especially STAT1, STAT3, and TLR-4.

### 2.5. LCN2 Decreased UPEC Infections in Bladder Cells, Which May Be Associated with the Regulation of the JAK/STAT Signaling Pathway

Our results show that LCN2 might decrease UPEC infections in bladder epithelial cells by regulating the JAK/STAT signaling pathway in a high-glucose environment. To further confirm that LCN2 suppressed UPEC infections in bladder cells through this signaling pathway, SV-HUV-1 cells were co-incubated with either the JAK inhibitor Ruxolitinib or the STAT inhibitor Fludarabine (25 and 50 mM/mL, respectively) with glucose to block the pathway, followed by LCN2 pretreatment and infecting the cells with UPEC as described. As shown in Figure 7, the treatment of bladder cells with LCN2 significantly reduced glucose-enhanced UPEC infectivity after 24 h co-pretreatment with glucose and with or without JAK or STAT inhibitors. (*p* < 0.001 compared to the 15 mM glucose-treated group). On the other hand, pretreatment with JAK or STAT inhibitors prior to the LCN2 treatment showed only a slight, but not statistically significant, increase in UPEC infection in the high-glucose environment. However, in the absence of LCN2 treatment, UPEC infection rates were significantly higher in the groups pretreated with JAK or STAT inhibitor alone than in cells pretreated with both an inhibitor and LCN2 (*p* < 0.01 and *p* < 0.05, respectively, for JAK and STAT inhibitor pretreatments compared to each other’s LCN2 plus groups). Thus, the results suggest that the inhibitory effect of LCN2 may not only be regulated by the JAK/STAT pathway.

### 2.6. Increased UPEC Infections in Bladder Cells in a High-Glucose Environment Were Inhibited by LCN2 Downregulating the JAK/STAT-Pathway-Related Proteins

Figure 8 shows the expressions of IFN-γ, IL-6, TLR4, and JAK/STAT-path-way-related proteins in SV-HUC-1 cells that were co-incubated with either JAK or STAT inhibitors before LCN2 pretreatment with UPEC infection. The results revealed that there was a significant rebound in the expression of JAK2 but not JAK1 in the group with STAT inhibitor co-incubation, regardless of LCN2 pretreatment; whereas STAT1 and STAT3 levels increased after STAT inhibitor and LCN2 co-incubation and STAT inhibitor pretreatment alone, respectively (no significant difference compared to the 15 mM glucose-pretreated group). However, all pSTAT1/3 and the inflammatory factors IL-6 and TLR-4 exhibited rebounds after co-incubated with the JAK inhibitor in UPEC-infected bladder cells with or without LCN2 pretreatment, especially pSTAT1 and IL-6 (no significant difference or higher compared to the 15 mM glucose-pretreated group, *p* < 0.001). In addition, SOCS3 in either the JAK or STAT inhibitor co-incubated groups, whether with/without LCN2 pretreatment, showed no significant difference in expression levels compared to the 15 mM glucose-pretreated group. In addition, IFN-γ in either the JAK or STAT inhibitor co-incubated groups, whether with/without LCN2 pretreatment, was significantly lower than the 15 mM glucose-pretreated group (*p* < 0.0001). The results of the LCN2 pretreatment showed that both JAK and STAT have regulatory expressions that inhibit rebounds from each other. The STAT inhibitor pretreatment increased the expression of JAK2, while the effect of the JAK inhibitor significantly raised pSTAT1/3 expression levels. However, this also indicates that the anti-inflammatory and UPEC-inhibiting effects of LCN2 are not entirely regulated by the JAK/STAT pathway.

## 3. Discussion

Our laboratory has verified that sugar enhances UPEC-induced bladder cell infection and inflammation by activating TLR-4- and JAK/STAT1-dependent pathways [26]. We recently demonstrated that insulin could reduce JAK/STAT-pathway-regulated UPEC infections in bladder cells in a high-glucose environment [20]. However, in comparison with insulin, which mainly acts on blood sugar stabilization and diabetes treatment, more effective biological drugs are warranted to improve UPEC infection in the urinary tract in a high-glucose environment. Therefore, in the present study, we found that the AMP LCN2 could more effectively downregulate the JAK/STAT signaling pathway in a high-glucose environment and inhibit UPEC infection in bladder epithelial cells.

The expression levels of virulence genes in UPEC were significantly downregulated in the presence of insulin and/or glucose [29]. As DM is associated with insufficient insulin production and action, insulin may regulate AMP production. AMP production is induced by the activation of the insulin-mediated PI3K/AKT pathway, eventually protecting urothelial cells from the invading UPEC [15]. We previously indicated that STAT1 and STAT3 in the JAK/STAT pathway play critical roles in the regulation of UPEC infection and colonization in uroepithelial cells, such as prostate or bladder cells, in high-glucose environments [26,27,30]. Therefore, it is necessary to understand the synergistic functions of AMP effects and JAK/STAT signaling pathway regulation in UPEC-infected bladder cells. As shown in Figure 1, LCN2 effectively reduced UPEC infection in bladder cells in a dose-dependent manner. SV-HUC-1 cells pretreated with 5 or 25 μg/mL LCN2 showed significantly decreased UPEC colonized numbers compared to the cells treated with 15 mM glucose. The infection rate in the 25 μg/mL LCN2-pretreated group was even lower than that in the general UPEC-infected group. LCN2 pretreatment not only reduced UPEC infection within the bladder cells but also suppressed STAT1/3 and TLR-4 expression in a dose-dependent manner. The expression of STAT1 and STAT3 protein in SV-HUC-1 cells was synchronously downregulated with the UPEC infection along with TLR-4 expression (Figure 2, Figure 3 and Figure 4). The regulatory effect of STAT1 appears to be more pronounced than that of STAT3, as STAT1 was downregulated upon pretreatment with 5 μg/mL LCN2. LCN2 exerts selective pressure on bacterial growth in the bladder. In a human cohort of women with recurrent *E. coli* UTIs, urine LCN2 levels were associated with UTI episodes and bacteriuria levels that were also related to the number of siderophore systems [31]. Another study showed that the ileum of Lcn2-null mice displayed an abundance of segmented filamentous bacteria and caused persistent colonization, provoking epithelial antimicrobial responses and affecting T-helper-cell polarization [32]. These findings reveal the importance of LCN2 in uroepithelial innate immunity and its good antibacterial ability.

Wang et al. found that the microRNA miR-383 may suppress LCN2 expression and disrupt the activation of the JAK/STAT signaling pathway to reduce keratinocyte proliferation and induce apoptosis in psoriasis progression [33]. Furthermore, LCN2 plays a role in β-cell function in inflammatory conditions. STAT1 activation has been highlighted to facilitate the upregulation in LCN2 expression in pancreatic islet β cells [34]. Therefore, the relationship between LCN2 production and the JAK/STAT regulatory pathway is worth exploring to reveal the mechanism underlying UPEC infection in bladder epithelial cells. Our results clearly demonstrate that LCN2 can significantly inhibit the proteins related to the JAK/STAT signaling pathway (including JAK1/2, STAT1, STAT3, and their phosphorylated active forms) in bladder cells by downregulating the expression of TLR-4, thereby decreasing proinflammatory cytokines such as IFN-γ and IL-6. All related proteins showed very similar expression trends, especially in the 5 and 25 μg/mL LCN2-pretreated groups. SOCS protein, a JAK/STAT pathway inhibitor, can suppress proinflammatory responses in human uroepithelial cells during long-term UPEC infections [35,36]. Our results also showed that SOCS3 protein expression was negatively correlated with the LCN2 concentration and increased in the 25 μg/mL LCN2 treatment group, although the value was not significantly different from the 15 mM glucose treatment group. JAK/STAT signaling is a vital pathway in numerous developmental and homeostatic processes and regulates crucial inflammatory processes [37]. STAT1 and NF-κB may work together as a complex to control the upregulation in LCN2 expression in the retina and stimulate an inflammatory response, leading to increased infiltration of LCN2-positive neutrophils in the choroid and retina of early age-related macular degeneration [38]. In addition, in experiments with liver-injured mice, LCN2 expression increased after radiation exposure, tumor necrosis factor-α treatment, or both; STAT3 appeared to be involved in the signaling cascade [39]. In this study, we demonstrated a close relationship between LCN2 and the JAK/STAT signaling pathway in UPEC infection of bladder cells.

Our previous study revealed that inhibiting JAK/STAT signaling by testosterone treatment suppressed UPEC infections and intercellular bacteria community (IBC) formation in prostate cells [27,30]. However, another study of ours showed that the inhibition of UPEC infection in bladder cells by another AMP, RNase 7, was not completely perturbed by JAK or STAT inhibition when compared to the treatment groups with a JAK inhibitor or STAT inhibitor alone. In contrast, although the LCN2 treatment also showed that it was not fully affected by JAK or STAT inhibition, the interference effects of both inhibitors on protein expression were more significant than that of RNase 7 [28]. In addition, the results showed that, although only a partial effect, it indeed reduced the efficacy of LCN2 in inhibiting UPEC infections in bladder cells, as impeded by JAK and STAT inhibitors. Our results also showed that the inhibition of STAT/JAK on LCN2 would interact with each other to influence the expression of respective proteins. The inhibition of JAK led to the expression of pSTAT1, pSTAT3, TLR4, and IL-6, while blocking STAT led to the partial rebound expression of STAT1, STAT3, and JAK2. The present study demonstrates that blocking JAK/STAT signaling in a high-glucose environment does not totally affect the LCN2-induced reduction in UPEC infections in bladder cells. Conversely, JAK inhibition is relatively important in the inflammatory regulation of RNase 7 pretreatment. The inhibition of UPEC infection in bladder cells by LCN2 may be affected by both JAK and STAT regulation. Therefore, this study further showed that the downregulation of inflammation in bladder cells infected with UPEC by different AMPs is also not completely consistent, even though they may both be regulated through the JAK/STAT pathway. Whether this is due to the different properties and structures of ribonuclease (RNase7) and lipocalin (LCN2) deserves further study.

## 4. Materials and Methods

### 4.1. Bacterial Strain

As a model, we used the UPEC strain CFT073 from the American Type Culture Collection (ATCC 700928) [40]. UPEC was transformed with the pGFP plasmid (Clontech, Palo Alto, CA, USA) to express green fluorescence. Bacterial strains were cultured in Luria–Bertani (LB) broth supplemented with ampicillin (100 µg/mL) and incubated at 37 °C overnight. Bacterial growth was spectrophotometrically determined at an optical density of 600 nm (OD_600_). For in vitro infections, bacteria were suspended in culture medium at a multiplicity of infection (MOI) of 1:100.

### 4.2. Bladder Cell Culture, LCN2 Pretreatment, and UPEC Infection

The human normal bladder cell line SV-HUC-1 was cultured in F-12K medium (Kaighn’s Modification of Ham’s F-12 Medium) containing 2 mM L-glutamine and 1500 mg/L sodium bicarbonate (GIBCO-BRL #21127-022) in the presence of 50 ng/mL bovine pituitary extract medium at 37 °C in a 5% CO_2_ incubator. The culture medium was replaced every 2–3 days. A cell culture medium containing 15 mM glucose was used to culture 5.5 × 10^5^ cells for 24 h; the cells were then treated with LCN2 (MyBioSource, San Diego, CA, USA) at different concentrations (1, 5, and 25 μg/mL) for 24 h and infected with pGFP-UPEC at an MOI of 1:100 for 4 h. Finally, the bacterial adhesion and invasion abilities were detected by immunofluorescence microscopy and the plate counting method, respectively. Untreated cells were used as the normal control group, and cells only infected with pGFP-UPEC were used as the positive control group. The other group of cells undergoing the same treatment were mixed with RIPA buffer, and the cell protein was collected for subsequent protein expression determination. The signal blocking experiments were executed by pretreatment with 25 μM JAK inhibitor (Ruxolitinib; MedChem Express, Princeton, NJ, USA) or 50 μM STAT inhibitor (Fludarabine; Selleck Chemicals Inc., Houston, TX, USA) with 15 mM glucose for 24 h, followed by the LCN2 pretreatment (25 μg/mL), UPEC infection, and subsequent experiments, as describe above.

### 4.3. UPEC Infection and Quantification by Colony-Forming Unit (CFU) Counting and Fluorescence Microscopy

After a 4 h UPEC incubation period, the infected monolayers were washed four times with phosphate-buffered saline (PBS) and incubated for 30 min in a growth medium containing gentamicin (100 μg/mL; Sigma-Aldrich, St. Louis, MO, USA). To measure bacterial invasion, the cells were lysed and harvested using 0.5% trypsin (Gibco; Invitrogen, Grand Island, NY, USA) and 0.1% Triton X-100 (Amresco, Solon, OH, USA) then plated onto a nutrient broth (NB) medium containing ampicillin. The total CFUs were counted after 24 h of incubation to quantify bound bacteria. To detect the green fluorescence of UPEC, SV-HUC-1 cells seeded onto 18 mm coverslips were infected with pGFP-UPEC (MOI of 100) and observed by fluorescence microscopy, as previously described [41]. Human anti-STAT1, STAT3, and TLR-4 phycoerythrin (PE)-conjugated antibodies (R&D Systems; Minneapolis, MN, USA) were used. At least three coverslips per condition were examined. To quantify invasion, images of 20 random fields of each coverslip were acquired and counted using image analysis software (ImageJ software) (version 1.41o) (National Institutes of Health, Bethesda, MD, USA) (http://rsb.info.nih.gov/ij/, accessed on 2 October 2012).

### 4.4. Cell Protein Extraction

The treated cells were centrifuged at 500× *g* for 5 min at 4 °C, the supernatant was poured out, and the cell pellet was placed in a protein lysis buffer (Pierce, Rockford, IL, USA). The pellet was homogenized with a microgrinder and evenly mixed to collect the expressed proteins. After all the samples were collected, they were thawed on ice and centrifuged at 12,000 rpm at 4 °C for 10 min. The supernatant was collected to obtain the total protein. The samples were stored in a refrigerator at −80 °C for later use.

### 4.5. Western Blot Analysis

The proteins in the samples were harvested using RIPA lysis buffer (Millipore; Burlington, MA, USA). The experimental procedure was conducted as per our previous publication [29]. The antibodies used were as follows: anti-JAK1 (BD Biosciences; San Diego, CA, USA), anti-JAK2, anti-STAT1 (Cell Signaling, Beverly, MA, USA), anti-pSTAT1, anti-STAT3 (Cell Signaling, Beverly, MA, USA), anti-pSTAT3, anti-suppressor of cytokine signaling 3 (SOCS3, Abcam; Cambridge, UK), anti-interferon (IFN)-γ (Abcam), anti-TLR-4 (Proteintech; Rosemont, IL, USA), anti-interleukin (IL)-6 (Bioworld Technology; Bloomington, MN, USA), and anti-β-actin (Santa Cruz; Dallas, TX, USA) at room temperature (~25 °C) for 1 h. After incubation with an appropriate secondary horseradish-peroxidase-conjugated IgG antibody (R&D Systems; Minneapolis, MN, USA) for 30 min at room temperature, the protein bands on the membrane were detected using an ECL-Plus Western Blot Detection system (GE Healthcare UK Ltd.; Buckinghamshire, UK) according to the manufacturer’s instructions. All experiments were performed at least three times. A graphical analysis of band density was performed using ImageJ software (version 1.41o) (National Institutes of Health, Bethesda, MD, USA) (http://rsb.info.nih.gov/ij/, accessed on 2 October 2012).

### 4.6. Cytometric Bead Array (CBA) Immunoassay

The cell culture medium was centrifuged (13,000× *g* for 20 min at 4 °C), and the supernatant was assessed using a human inflammatory cytokine CBA (BD Biosciences, San Diego, CA, USA) for the cytokines IL-8, IL-10, IL-1β, and IL-6. The cytokine capture bead, PE detection reagent, and recombinant standards or test samples were incubated for 3 h at room temperature. An FACSCanto flow cytometer (BD Biosciences, San Diego, CA, USA) was used to acquire the data, which were analyzed using the BD CBA analysis software to produce graphs [42,43].

### 4.7. Statistical Analysis

Data are expressed as means ± standard deviations (SDs). An analysis of variance was used to evaluate differences between various treatment groups and controls. Statistical differences between groups were determined using Student’s *t*-test. Statistical significance was set at *p* < 0.05.

## 5. Conclusions

Taken together, our study provides evidence that the AMP LCN2 can reduce UPEC infections in bladder epithelial cells by decreasing the activation of the JAK/STAT signaling pathway in a high-glucose environment. The preventive function of LCN2 against UPEC infection is dose-dependent and associated with TLR-4 expression. These findings may provide a rationale for targeting LCN2/TLR-4/JAK/STAT regulation in bacterial cystitis treatment. Nevertheless, further studies are imperative to explore the specific mechanism by which the LCN2, TLR-4, and JAK/STAT pathways participate in UPEC-induced inflammation, with the goal of developing more effective therapies for cystitis.

## Figures and Tables

**Figure 1 ijms-23-15763-f001:**
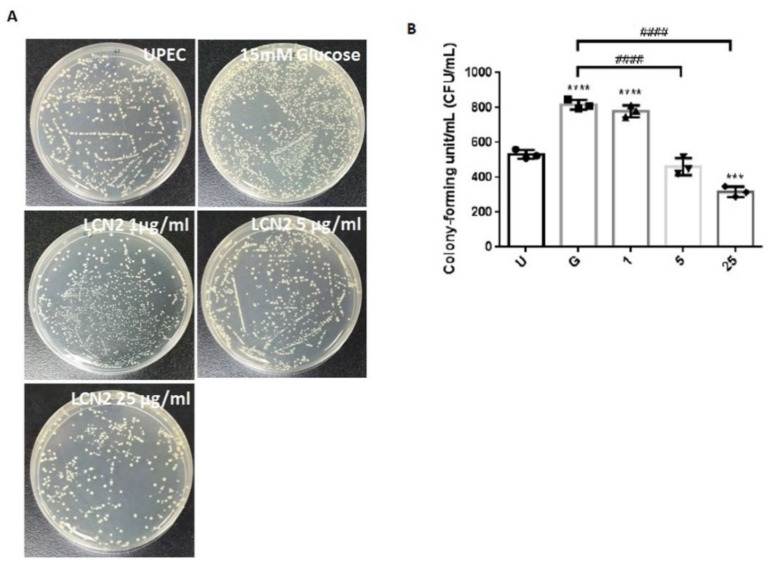
LCN2 suppression of UPEC infections in bladder cells in a high-glucose environment. SV-HUC-1 cells were first pretreated with 15 mM glucose, followed by treatments with different concentrations of LCN2 (1, 5, and 25 μg/mL) for 24 h and infection with pGFP-UPEC (MOI of 1:100) for 4 h, as described in the text. The definitions of the letters on the X axis: U, UPEC infection alone; G, with 15 mM glucose pretreatment alone. UPEC infections in bladder cells were examined by plating cells on NB agar (**A**) and were determined using ImageJ software (**B**). Colony-forming units/mL (CFUs/mL) were acquired after plating out lysed solutions of infected cells. The picture shown is representative of a typical result. The data are expressed as the means ± SDs of three independent experiments. An analysis of variance was used to evaluate differences between various treatment groups and controls. Statistical differences between groups were determined using the Mann–Whitney U Student’s *t*-test. Statistical significance was set at *p* < 0.05. *** *p* < 0.001, **** *p* < 0.0001 compared to the positive control groups. #### *p* < 0.0001 for comparisons between two groups.

**Figure 2 ijms-23-15763-f002:**
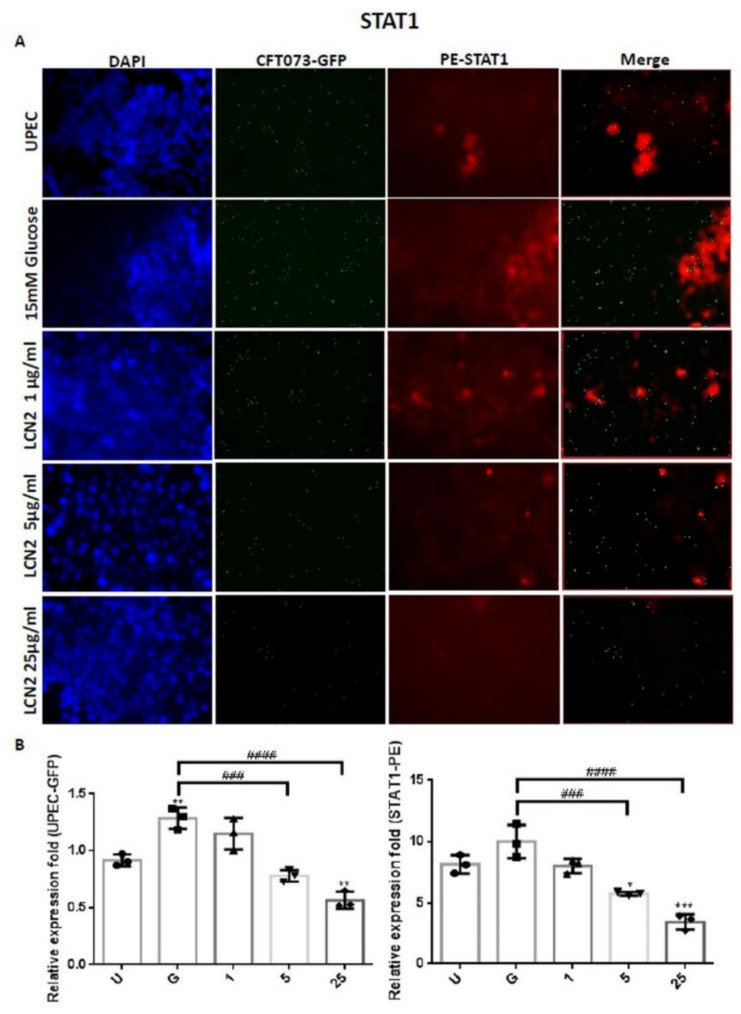
LCN2 decreased UPEC infections and STAT1 expression in bladder cells. SV-HUC-1 cells were first pretreated with 15 mM glucose, followed by treatment with different concentrations of LCN2 (1, 5, and 25 μg/mL) for 24 h and infection with pGFP-UPEC (MOI of 1:100) for 4 h, as described in the text. The definitions of the letters on the X axis: U, UPEC infection alone; G, with 15 mM glucose pretreatment alone. GFP-UPEC and PE-STAT1 expression in SV-HUC-1 cells, for 4 h postinfection, were (**A**) observed by fluorescence microscopy and (**B**) measured using ImageJ software. Cells infected with UPEC alone were used as a positive control. The pictures shown are representative of typical results. DAPI was used to count the number of cells and as a standard for calculating the fluorescence expression ratios of cells. Cell images were captured using a microscope (Leica) at 100× magnification. The data are expressed as means ± SDs from three separate experiments. An analysis of variance was used to evaluate differences between various treatment groups and controls. Statistical differences between groups were determined using the Mann–Whitney U Student’s *t*-test. Statistical significance was set at * *p* < 0.05. ** *p* < 0.01, *** *p* < 0.001 compared to the positive control groups. ### *p* < 0.001, #### *p* < 0.0001 for comparisons between two groups.

**Figure 3 ijms-23-15763-f003:**
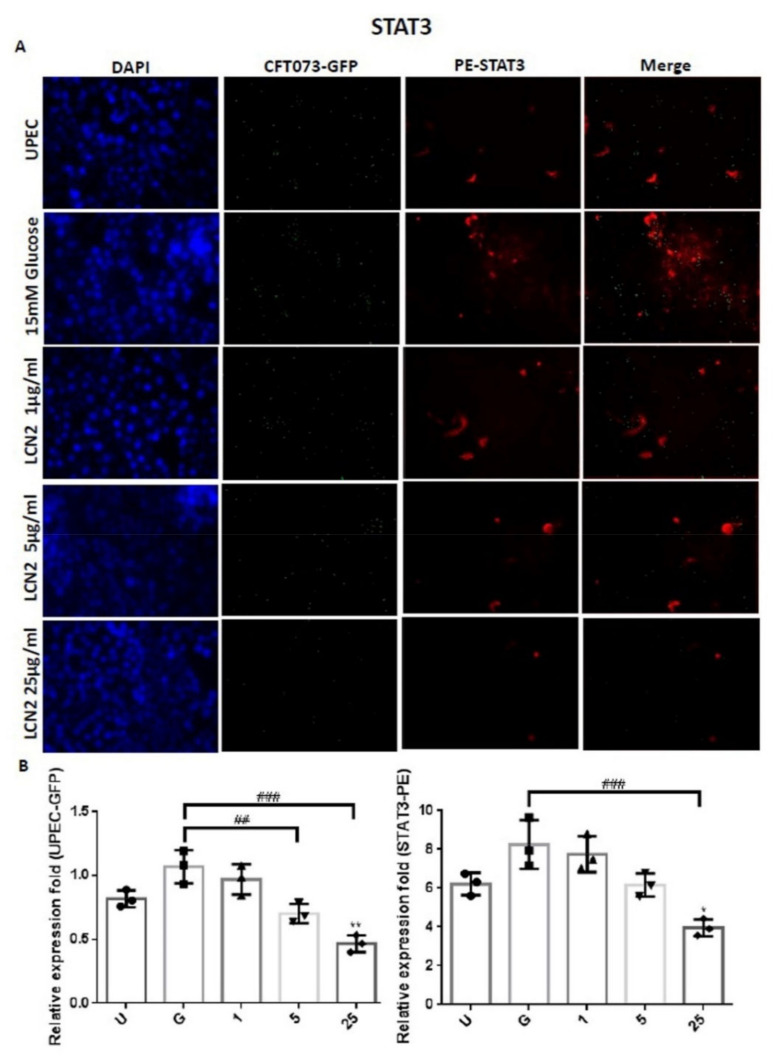
LCN2 decreased UPEC infections and STAT3 expression in bladder cells. SV-HUC-1 cells were first pretreated with 15 mM glucose, followed by treatment with different concentrations of LCN2 (1, 5, and 25 μg/mL) for 24 h and infection with pGFP-UPEC (MOI of 1:100) for 4 h, as described in the text. The definitions of the letters on the X axis: U, UPEC infection alone; G, with 15 mM glucose pretreatment alone. GFP-UPEC and PE-STAT3 expression in SV-HUC-1 cells, for 4 h postinfection, were (**A**) observed by fluorescence microscopy and (**B**) measured using ImageJ software. Cells infected with UPEC alone were used as positive controls. The pictures shown are representative of typical results. DAPI was used to count the number of cells and as a standard for calculating the fluorescence expression ratios of cells. Cell images were captured using a microscope (Leica) at 100× magnification. The data are expressed as means ± SDs from three separate experiments. An analysis of variance was used to evaluate differences between various treatment groups and controls. Statistical differences between groups were determined using the Mann–Whitney U Student’s *t*-test. Statistical significance was set at *p* < 0.05. * *p* < 0.05, ** *p* < 0.01 compared to the positive control groups. ## *p* < 0.01, ### *p* < 0.001 for comparisons between two groups.

**Figure 4 ijms-23-15763-f004:**
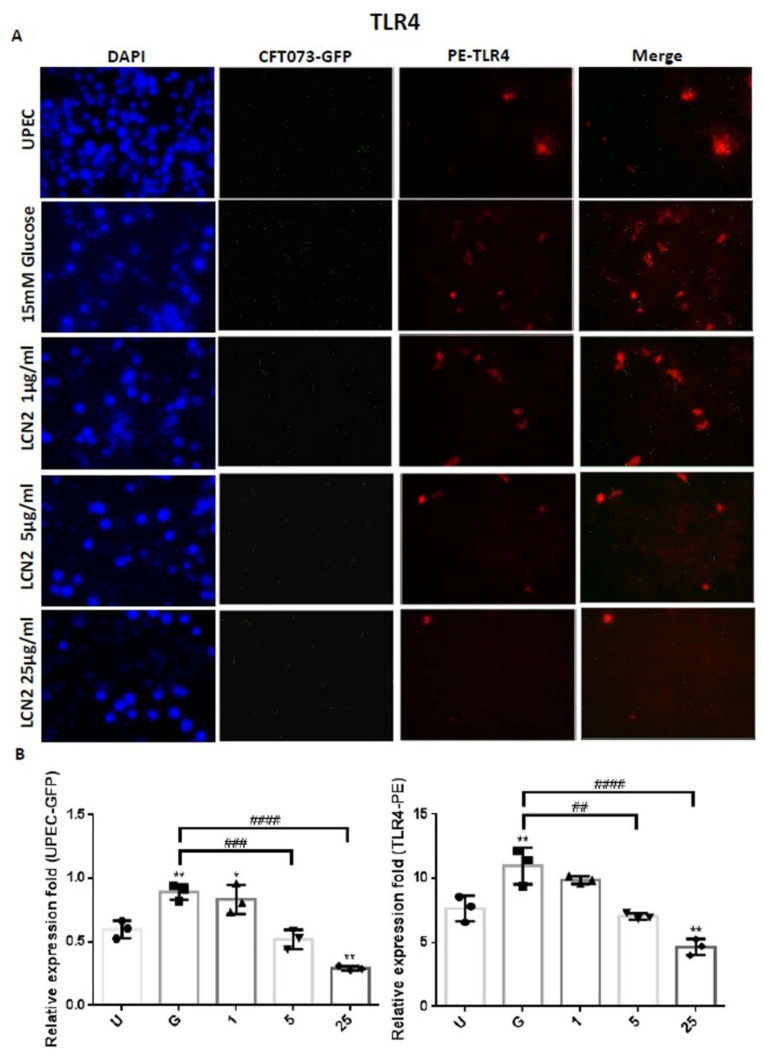
LCN2 decreased UPEC infections and TLR-4 expression in bladder cells. SV-HUC-1 cells were first pretreated with 15 mM glucose, followed by treatment with different concentrations of LCN2 (1, 5, and 25 μg/mL) for 24 h and infection with pGFP-UPEC (MOI of 1:100) for 4 h, as described in the text. The definitions of the letters on the X axis: U, UPEC infection alone; G, with 15 mM glucose pretreatment alone. GFP-UPEC and PE-TLR-4 expression in SV-HUC-1 cells, for 4 h postinfection, were (**A**) observed by fluorescence microscopy and (**B**) measured using ImageJ software. Cells infected with UPEC alone were used as positive controls. The pictures shown are representative of typical results. DAPI was used to count the number of cells and as a standard for calculating the fluorescence expression ratios of cells. Cell images were captured using a microscope (Leica) at 100× magnification. The data are expressed as means ± SDs from three separate experiments. An analysis of variance was used to evaluate differences between various treatment groups and controls. Statistical differences between groups were determined using the Mann–Whitney U Student’s *t*-test. Statistical significance was set at *p* < 0.05. * *p* < 0.05, ** *p* < 0.01 compared to the positive control groups. ## *p* < 0.01, ### *p* < 0.001, #### *p* < 0.0001 for comparisons between two groups.

**Figure 5 ijms-23-15763-f005:**
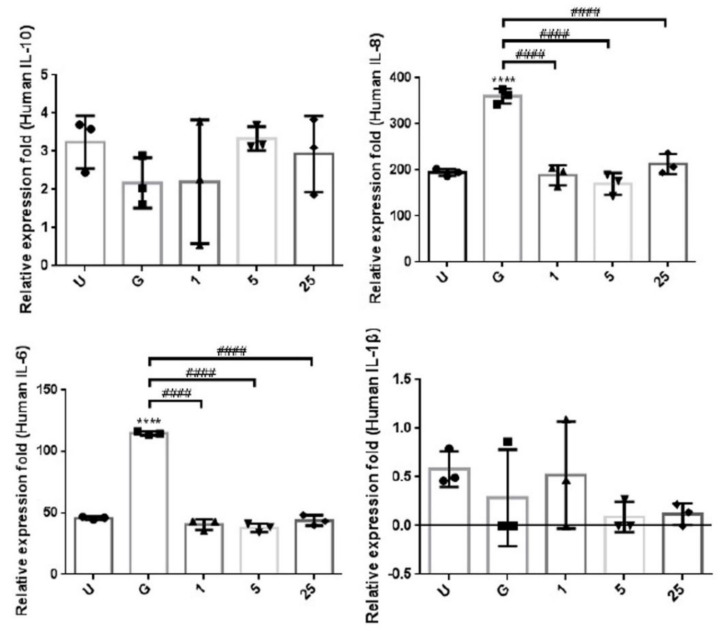
LCN2 suppressed the secretion of inflammatory cytokines in UPEC-infected bladder cells. SV-HUC-1 cells were first pretreated with 15 mM glucose, followed by treatment with different concentrations of LCN2 (1, 5, and 25 μg/mL) for 24 h and infection with pGFP-UPEC (MOI of 1:100) for 4 h, as described in the text. The definitions of the letters on the X axis: U, UPEC infection alone; G, with 15 mM glucose pretreatment alone. The release levels of cytokines IL-8, IL-10, IL-1β, and IL-6 from UPEC-infected SV-HUC-1 cells were measured by CBA. Cells infected with UPEC alone (MOI of 1:100) were used as positive controls. The data are expressed as means ± SDs from three separate experiments. An analysis of variance was used to evaluate differences between various treatment groups and controls. Statistical differences between groups were determined using the Mann–Whitney U Student’s *t*-test. Statistical significance was set at *p* < 0.05. **** *p* < 0.0001 compared to the positive control groups. #### *p* < 0.0001 for comparisons between two groups.

**Figure 6 ijms-23-15763-f006:**
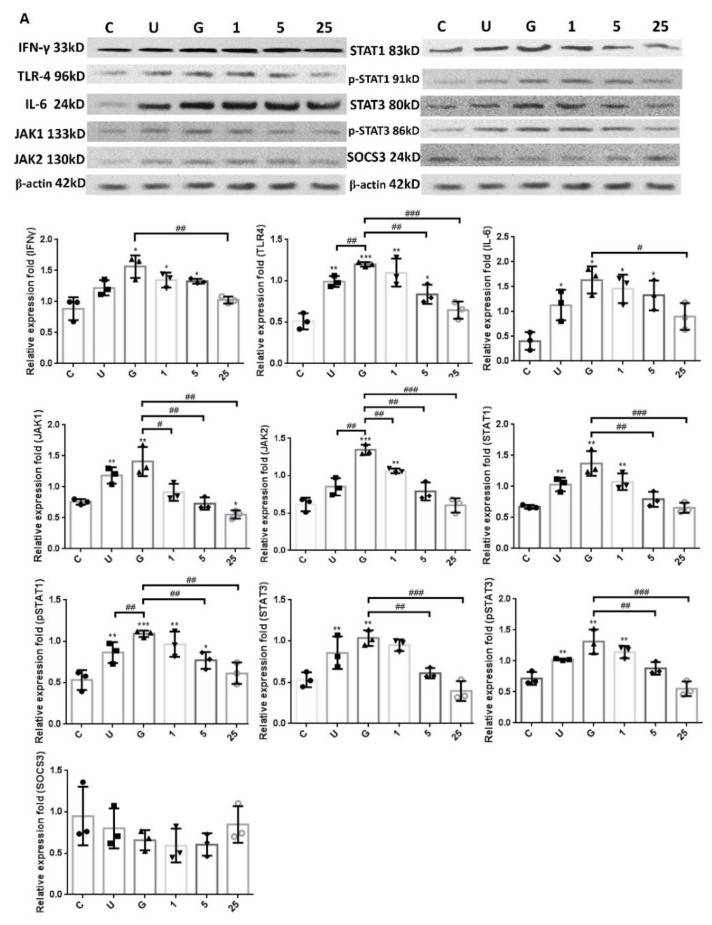
LCN2 downregulated the expression of protein associated with the JAK/STAT pathway and inflammatory responses in UPEC-infected bladder cells in a high-glucose environment. SV-HUC-1 cells were first pretreated with 15 mM glucose, followed by treatment with different concentrations of LCN2 (1, 5, and 25 μg/mL) and 15 mM glucose for 24 h and infection with pGFP-UPEC (MOI of 1:100) for 4 h, as described in the text. The definitions of the letters on the X axis: C, negative control; U, UPEC infection alone; G, with 15 mM glucose pretreatment alone. The total protein from all cell groups was collected for a test. (**A**) Total protein expressions of JAK1, JAK2, STAT1, STAT3, pSTAT1/STAT3, SOCS3, TLR-4, IL-6, and IFN-γ in cells were analyzed by Western blotting. The picture shown is representative of a typical result. (**B**) Expression results were assessed using a densitometer and quantified using ImageJ software (NIH). The results are presented as means ± SDs of three independent experiments. All data were normalized by their own internal reference, β-actin. An analysis of variance was used to evaluate differences between various treatment groups and controls. Statistical differences between groups were determined using the Mann–Whitney U Student’s *t*-test. Statistical significance was set at *p* < 0.05. * *p* < 0.05, ** *p* < 0.01, *** *p* < 0.001 compared with the respective control groups. # *p* < 0.05, ## *p* < 0.01, ### *p* < 0.001 for comparisons between two groups.

**Figure 7 ijms-23-15763-f007:**
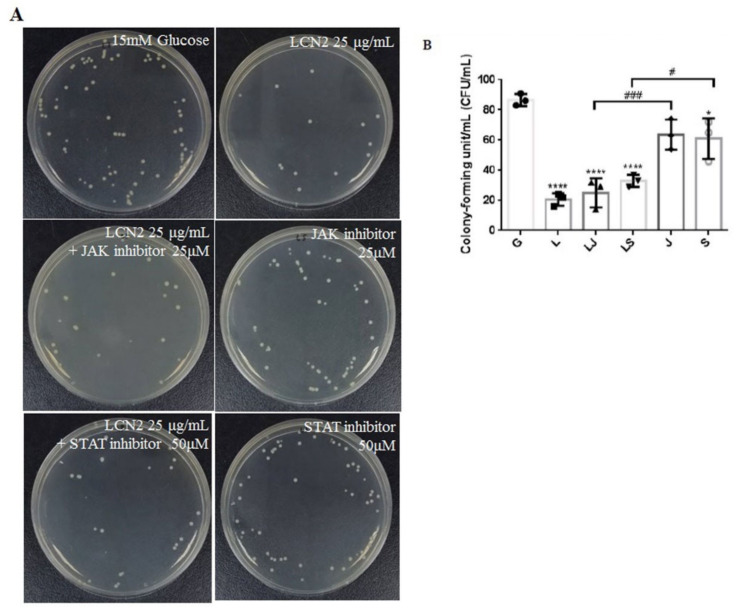
JAK and STAT inhibitors do not affect LCN2 suppression in UPEC-infected bladder cells in a high-glucose environment. SV-HUC-1 cells were pretreated with a 25 μM JAK inhibitor (LJ) or a 50 μM STAT inhibitor (LS) and 15 mM glucose for 24 h before a 25 μg/mL LCN2 pretreatment followed by UPEC infection (MOI of 100), as described above. Cells pretreated with the 25 μM JAK inhibitor (J) or STAT inhibitor (S) alone for 24 h followed by UPEC infection were used as respective controls. Cells pretreated with 25 μg/mL LCN2 alone (L) followed by UPEC infection were used as other controls. Cells infected with 15 mM glucose and UPEC alone were used as the respective positive controls (G). The picture shown is representative of a typical result. Twenty-four hours postinfection, all infected cells were (**A**) lysed and plated on LB agar and (**B**) measured using ImageJ software to analyze UPEC colonization. Colony-forming units/ml (CFUs/ml) were acquired after plating 10-fold dilutions of infected cells with lysis. The data are expressed as means ± SDs of three separate experiments. An analysis of variance was used to evaluate differences between various treatment groups and controls. Statistical differences between groups were determined using the Mann–Whitney U Student’s *t*-test. Statistical significance was set at *p* < 0.05. * *p* < 0.05, **** *p* <0.0001 compared to the positive control groups. # *p* < 0.05, ### *p* < 0.001 for comparisons between the two indicated groups.

**Figure 8 ijms-23-15763-f008:**
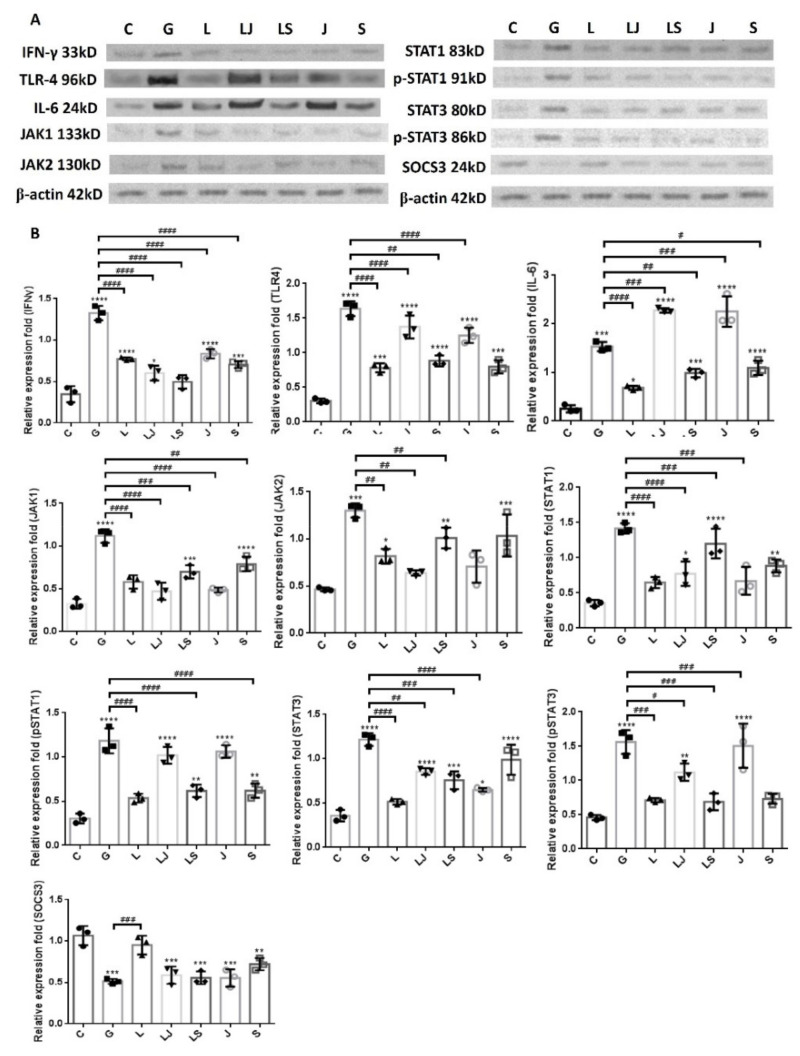
JAK and STAT inhibitors affected the LCN2-downregulated JAK/STAT pathway and inflammatory factors in UPEC-infected bladder cells in a high-glucose environment. SV-HUC-1 cells were pretreated with a 25 μM JAK inhibitor (LJ) or 50 μM STAT inhibitor (LS) and 15 mM glucose for 24 h before a 25 μg/mL LCN2 pretreatment followed by UPEC infection (MOI of 100), as described above. Cells pretreated with the 25 μM JAK inhibitor (J) or STAT inhibitor (S) alone for 24 h followed by UPEC infection were used as respective controls. Cells pretreated with 25 μg/mL LCN2 alone (L) followed by UPEC infection were used as other controls. Cells infected with 15 mM glucose and UPEC alone were used as the respective positive controls (G). The picture shown is representative of a typical result. The total protein from all cell groups was collected for detection. (**A**) The total protein expression of JAK1, JAK2, STAT1, STAT3, phosphorylated-STAT1/STAT3, the inhibitor SOCS3, TLR4, IL-6, and IFN-γ were analyzed using Western blotting. (**B**) All data were normalized by their own internal reference, β-actin. The results were assessed using a densitometer and quantified using ImageJ software (NIH). The results are presented as means ± SDs of three independent experiments. * *p* < 0.05, ** *p* < 0.01, *** *p* < 0.001, **** *p* <0.0001 compared to the respective control groups. An analysis of variance was used to evaluate differences between various treatment groups and controls. Statistical differences between groups were determined using the Mann–Whitney U Student’s *t*-test. Statistical significance was set at *p* < 0.05. # *p* < 0.05, ## *p* < 0.01, ### *p* < 0.001, #### *p* <0.0001 for comparisons between the two individual groups.

## Data Availability

Not applicable.

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
