# Peer review of "Antimicrobial Peptide LCN2 Inhibited Uropathogenic *Escherichia coli* Infection in Bladder Cells in a High-Glucose Environment through JAK/STAT Signaling Pathway"

_ijms, 2022, doi:10.3390/ijms232415763_

Round 1
Reviewer 1 Report
I would like to thank the authors for such an interesting study. Please see the following for some minor suggestions:
1. line 38: should define DM as it first appeared in the introduction.
2. line 102, it seems like there was an extra "I" at the end of the line.
3. line 103-105, the sentence is a little awkward, may I suggest the authors reword the sentence to clarify your idea?
4. Figure 1
- Fig 1B. CFU units should be CFU/mL. The abbreviations (U, G) in the graph should be explained in the legend, and LCN2 should be somewhere on the x-axis.
- legend: The method provided in the legend was unclear. "SV-120 HUC-1 cells were pretreated with different concentrations of LCN2 (1, 5, and 25 μg/mL) and 15 mM glucose for 24 h and then infected with pGFP-UPEC (MOI:1:100)": When reading this sentence alone, one might wonder why the pretreatment of LCN2 was compared to 15mM glucose. May I suggest that the authors make it clear that all conditions are pretreated with 15mM glucose, followed by treatments of LCN2, and then infected by pGFP-UPEC. This confusion can also be avoided by using a "vehicle control" instead of comparing LCN2 conditions to "glucose".
5. Similar comments as above for Figures 2, 3, and 4
6. line 184-185. It's unclear what is the downward trend referring to
Author Response
Reviewer1
Comments and Suggestions for Authors
I would like to thank the authors for such an interesting study. Please see the following for some minor suggestions:
- line 38: should define DM as it first appeared in the introduction.
A: Thanks for the reviewer's comment, we have corrected the description in the introduction. (Line 38)
- line 102, it seems like there was an extra "I" at the end of the line.
A: Thanks for the reviewer's comment, we have fixed the mistake. (Line 103)
- line 103-105, the sentence is a little awkward, may I suggest the authors reword the sentence to clarify your idea?
A: Thanks for the reviewer's suggestion, we have revised the description of the sentence to make the meaning clearer. (Line 104-106)
- Figure 1
- Fig 1B. CFU units should be CFU/mL. The abbreviations (U, G) in the graph should be explained in the legend, and LCN2 should be somewhere on the x-axis.
- legend: The method provided in the legend was unclear. "SV-120 HUC-1 cells were pretreated with different concentrations of LCN2 (1, 5, and 25 μg/mL) and 15 mM glucose for 24 h and then infected with pGFP-UPEC (MOI:1:100)": When reading this sentence alone, one might wonder why the pretreatment of LCN2 was compared to 15mM glucose. May I suggest that the authors make it clear that all conditions are pretreated with 15mM glucose, followed by treatments of LCN2, and then infected by pGFP-UPEC. This confusion can also be avoided by using a "vehicle control" instead of comparing LCN2 conditions to "glucose".
A: Thanks for the reviewer's comment, we are sorry for the confusing description in the figure legends. We have revised the related description in the legend as the reviewer’s suggestion. Hope these revision could make the meaning of figure legend clearer. (Line 127-130; line 132; line 285; Fig.1; Fig.7)
- Similar comments as above for Figures 2, 3, and 4
A: Thanks for the reviewer's comment, we have revised the related description in all the legend as the reviewer’s suggestion. Hope these revision could make the meaning of figure legend clearer. (Line 155-159; line 169-173; line 183-187; line 210-215; line 242-247)
- line 184-185. It's unclear what is the downward trend referring to
A: Thanks for the reviewer's comment, we have revised the description in this section and deleted the sentence about “downward trend” to avoid misunderstanding. (Line 203-205)

Reviewer 2 Report
1. Page 1, Line 38: Please claim what DM stands for.
2. Page 10, Line 23s: Please specific what JAK and STAT inhibitors were used. Why 25 and 50 uM inhibitor was used in the experiment? What is the EC50 or IC50 of those inhibitor?
3. In dicussion, the reason why the inhibitor and LCN2 combination is worse than LCN2 only needs to be discussed.
Author Response
Reviewer2
Comments and Suggestions for Authors
- Page 1, Line 38: Please claim what DM stands for.
A: Thanks for the reviewer's comment, we have corrected the description in the introduction. (Line 38)
- Page 10, Line 23s: Please specific what JAK and STAT inhibitors were used. Why 25 and 50 uM inhibitor was used in the experiment? What is the EC50 or IC50 of those inhibitor?
A: Thanks for the reviewer's comment, we have supplemented the names of JAK and STAT inhibitor in the section. Related description also highlighted in the Material section. (Line 260-261; line 436-438). In addition, the dose selection of JAK and STAT inhibitors is a continuation of our previous experiments, which have been confirmed to be the best effective doses. The related papers are supplemented below. Ruxolitinib EC50 values ranged from 25 to 50 uM and Fludarabine EC50 values ranged from 50 to 100 uM for inhibiting LCN2 according to SV-HUC-1 cell populations in our previous study. (Ref. 20; 26-28)
- In dicussion, the reason why the inhibitor and LCN2 combination is worse than LCN2 only needs to be discussed.
A: Thanks for the reviewer's comment. In the study we hypothesized that the efficacy of LCN2 in inhibiting UPEC infection of bladder cells would be regulated by JAK and STAT, and the results showed that although only a partial effect, it was indeed interfered, which explains the why the inhibitor and LCN2 combination is worse than LCN2 for UPEC inhibition. WB data also showed similar results. We have supplemented this part to highlight our opinion in the discussion section. (Line 397-400)
